# Insincerity, Secrecy, Neutralisation, Harm: Reporting Clergy Sexual Misconduct against Adults—A Survivor-Based Analysis

Stephen Edward de Weger

Faculty of Creative Industries, Education, and Social Justice, School of Justice, Queensland University of Technology, Brisbane, QLD 4000, Australia; stephen.deweger@qut.edu.au

**Abstract:** The foundational study for this article asked: how do survivors of clergy sexual misconduct against adults (CSMAA) in the Roman Catholic Church (RCC) describe and understand their experiences of reporting that misconduct to Roman Catholic Church authorities? The findings were that, while survivors sincerely believed that they would be cared for when they approached their Church officials, most soon began to sense a deep lack of insincerity coming from the officials dealing with their case. This insincerity was exposed in various forms by those officials as well as RCC hierarchy connected to the cases. The conclusion here is that the RCC seeks to neutralise exposure of CSMAA and the survivors thereof, and that they actually need to do so. The result—further and deeper harming of the already harmed.

**Keywords:** Roman Catholic; misconduct; abuse; adults; reporting; clericalism; power

## 1. Introduction

Clergy sexual misconduct or abuse against adults (CSMAA) causes serious and often life-long harms. However, for those who report CSMAA, another set of harms often overtakes and compounds those already endured. Following are the findings concerning the experiences of participants during and after official reporting process thereof (de Weger 2020)[1]. While the two geneses of harm—abuse and reporting—obviously interreact and connect, it is important to differentiate the harm of the initial abuse by individual clergy, from the latter 'harming of the harmed' when that abuse was reported. In short, the latter harm results when one's trust in an often lifelong-held, whole-of-life world view is betrayed. As such, the reporting experience, if it does not go well, can be even more devastating than the initial abuse-based harm. The reason for this is that this time, the harm experienced has been at the hands of the whole RCC community or faith family within which the survivor found their sense of meaning and belonging. Lack of official and unofficial validation by the victim's leaders and general community, often expressed as covert or overt victim-blaming, is spiritually and psychologically crushing. This lack of validation also enflames often-existing feelings of pain, confusion, guilt, and self-blame already often festering within victims because of the actual abuse, especially those who have secreted away their experiences because they are too fearful to disclose or report their abuses.

Reporting clergy sexual misconduct against adults to Roman Catholic Church authorities: An analysis of survivor perspectives (de Weger 2020) was an Australian-based study undertaken because there was a very large gap in the literature and research regarding CSMAA, and a complete vacuum of research into the reporting thereof. While this was definitely the case within Australia where the study was based, this lack of research is also a much broader one. In particular, the actual voices of victims/survivors of CSMAA, had rarely been brought to the fore. This project was also undertaken because it became clear in de Weger (2016) that many of the participants who had reported their abuses to RCC

officials had experienced negative responses, but those responses were not able to be dealt with as part of this first study[2].

The overall conclusion of both studies was that official church responses were deeply flawed due to erroneous beliefs about CSMAA, which in turn influenced the attitudes and behaviours towards survivors. However, RCC official processes had to be seen to be upholding Church law and teachings. As such, frontstage, the RCC needed to present a spirit of support for victims, and action against sexually active and misconducting clergy. Such tension between front and backstage realities, cannot help but produce cognitive dissonance and an overall flavour of insincerity. This insincerity was particularly revealed when survivors did not submissively comply with RCC processes offered, or how it was being offered. Much of this insincerity could be traced back to the RCC's own ambivalence about clergy sexual activity and misconduct, not to mention being caught off guard about that ambivalence (de Weger and Death 2017).

When the McCarrick case made headlines, ending with the Vatican's 2020 report on the case (Dulle 2020), it became clear that the RCC needed to face the growing dilemma as to how to respond to those reporting CSMAA. As a news report at the time stated:

> Since 2002, all bishops in the United States have known exactly how to address an allegation that a cleric sexually abused a child. The procedure is uniform and clear, and bishops seem to understand the importance of following it precisely and promptly. But the manner in which allegations of sexual misconduct with adults are handled looks nothing like those clear procedures.

> Church law does not expressly establish that sex between a cleric and an adult is a canonical crime. As a consequence, bishops everywhere find themselves vexed, and frequently, about how exactly they should handle allegations of clerical sexual misconduct involving adults—even in cases like McCarrick's, where coercion is an operative factor (Flynn 2018).

This vexedness was clearly evident in the testimonies of the participants in this study[3]. Furthermore, what emerged when reviewing the literature was, because of an already deeply tarnished reputation resulting from clergy child abuse, the RCC has an understandable vested interest in not allowing the reality of adult abuse to further contribute to an already collapsing reputation. What the McCarrick case and this study revealed also was that the Church has adopted similar processes to other elite and powerful institutions when faced with exposure of scandal: the deployment of a range of neutralisation tactics. These tactics and modes were first outlined in Anson Shupe's *In the Name of All That's Holy: A Theory of Clergy Malfeasance* (Shupe 1995). Shupe's neutralisation tactics were modified and expanded in de Weger (2020, pp. 229–45). As with most such institutions, the prime reason for the RCC resorting to such tactics is to maintain the secrecy of CSMAA and, perhaps even more so, to disguise the normalisation of more general clergy sexual activity.

This article has also been strongly influenced by Richard Sipe's 11-point thesis (Sipe 2008), which presents a somewhat practical and believable basis to CSMAA and the reasons for the need to cover-up clergy sexual activity and abuse in general. Sipe's thesis has been all but ignored by the RCC, but with the now many sordid stories of sexual abuse and misconduct within seminaries, against nuns, and against lay people (see Appendix 1, de Weger 2020), one could say Sipe's thesis is becoming somewhat prophetic, and increasingly so. The networks of secrecy and blackmail these abuses have created are also slowly emerging (e.g., Jesserer Smith 2019a, 2019b, 2019c; Niles 2021)[4].

According to Yocum (2013, pp. 93–107), the traditional projection of the celibate priesthood is one of a "symbolic system of purity" that, because of that proclaimed purity, holds great spiritual power and contains within it a divinely granted right to lead the flock. Yocum then relates this perception and institutional construct to the abuse crisis insofar as the power inherent in such a position, more easily enables abuse to occur, but also lies at the heart of why it has been so difficult to accept that 'holy' men could perpetrate such vile crimes (Yocum 2013, pp. 113–15). Shupe's more secular, sociological analysis of broader

elite institutional abuses explains that the existence of elite deviancy within a *religious* institution such as the RCC, should not at all be surprising. Shupe's whole theory espouses the broad, omnipresent expectation of deviancy occurring within powerful institutions. As Shupe (2007, p. 7) stated: we should be under no illusion but should

> expect "bad pastors" to be discovered as frequently as fleecing accountants, seducing professors, crooked cops, pilfering bankers, money-laundering corporate executive officers, and philandering therapists.

Deviancy within religious institutions should not exist . . . but it does. Why? According to Shupe, it is because of the iron law of oligarchy or in the case of religious institutions, the iron law of oligarchic clericalism (Shupe 1995, p. 56). Shupe explains how in every group of humans, there always rises a controlling elite. These elite have much power, and for many, such power can corrupt and become a tool for abuse. Such deviance and corruption then need to also be controlled and covered up so the elite can maintain their power of control, and their position.

In a strongly-worded letter written to Bishop McElroy in 2016, Sipe warned of the impact that continued ignoring of clergy sexual activity and misconduct would have on the RCC (Sipe 2016). The case of ex-Cardinal McCarrick was perhaps the first to fully publicly reveal the truth of high levels of sexual abuse within seminaries and between priests, and the part this activity played in control and of promotion or demotion of compromised clergy. With so much at stake there was then a subsequent constant need for cover-up of all forms of clergy abuse and even blackmail to maintain this secret culture of clerical deviancy so logically described in Sipe's 11-point thesis (Sipe 2008). With further investigations and reviewing of the literature dealing with CSMAA, of which there is precious little, it became clear that inadequate RCC responses to CSMAA may well be rooted in a subsequent deep fear of the exposure of the activities and controlling processes as described by Sipe. Indeed, there is a great deal at stake—the very central base of power and holiness of the RCC is its celibate/chaste, holy, chosen-by-God, male clergy (Doyle et al. 2006, pp. 8–9). Should the reality of the lack of adherence to these proclaimed teachings be exposed, the issue could well become the RCC's "feet of clay" (Old Testament: Book of Daniel 2: 33). Sipe complains at the end of his letter to Bishop McElroy: "I have tried to help the Church understand and heal the wounds of sexual abuse by clergy. My services have not been welcomed". However, his ignored insightful warnings are now being played out on the international stage.

## 2. Findings and Discussion

While a limitation of this study is its inability to generalise to the greater population, the nine participants in the study on which this article is based, is a number that fits within the requirements of a qualitative in-depth interview methodology. Furthermore, this study is validated and affirmed when triangulated with others such as Flynn (2003); Kennedy (2009); and Byrne (2010) not to mention the growing number of news stories concerning CSMAA and to how it is officially responded. To this can now be added a new study published during the writing of this present one, (Pooler and Barros-Lane 2022). This mixed method study of women survivors of what the authors term "clergy perpetrated sexual abuse of adults", also confirms many of the findings within de Weger (2020). While the study underpinning this article is focused on abuse within the Roman Catholic Church, when comparing findings from other denominations such as with the new Pooler and Barros-Lane (2022) study, what becomes clear is that there are underlying commonalities between denominations, commonalities which can also be found between religions, and indeed, between religious and secular institutions. Furthermore, it needs to be noted that in all the cases, the RCC Bishops, Archbishops, safeguarding leaders, and professionals that were involved are still in positions of authority and influence, and that the behaviours of these current RCC leaders and staff regarding those reporting CSMAA are serious and disturbing.

Rather than separating the findings and discussion thereof in two separate sections, it was more useful here to present these findings and discuss them within the framework of the tactics and modes of neutralisation to be presented. In many ways, all the findings and corresponding discussion in this article could be summed up in one specific experience: a confusing duality of intent accompanied by a disingenuous spirit of compassion and justice on the part of the officials to whom survivors went to report their CSMAA. In general, all participants expressed a sense of shock and sorrow resulting from unexpected reversals of positive relationships with clergy and officials. A perfect example of this was Adela's case.

Adela, emotionally and sexually groomed into a relationship in the 1970s by a well-known and highly respected spiritual director, author, and speaker, explained her first meeting with the offender's Religious Order representatives so poignantly:

> We were approaching [the Religious Order] as friends, but they turned us into enemies straight away. Almost, you know, instantly, we were enemies, even though we've been friends for years, and we were approaching them in that spirit.

An interesting and serendipitous element of de Weger (2020) was the broad time span of the nine survivor experiences both of CSMAA and the reporting thereof. Regarding the reporting, the time range was from the 1970s up to the 2020s. In Australia prior to 2000, reporting clergy abuse was a more informal and more parochial process. For Adela (and one other participant—Gary) there was still no formal reporting process for adult victims of clergy abuse. In the late 1990s, two separate official reporting procedures were created: Towards Healing and The Melbourne Response. However, these were both established to deal with child abuse[5]. Nevertheless, most victims of CSMAA did go through Towards Healing anyway[6]. It appears that within these RCC processes, such a reversal as described by Adela became less overt. Now, due to a lot of pressure to deal with the ever-exploding clergy abuse scandal, making enemies of those reporting such abuse could not be seen to be happening.

Regardless of the process or time, all the participants in this study at first felt little doubt that they were going to be believed, validated, and cared for. However, all soon began to sense a deep lack of sincerity on behalf of those they approached. What were originally seen as compassion-based invitations appeared to collapse into an often-shocking sense of antagonism or at least ambivalence towards complainants. This was especially the case if survivors started to voice confusion or discontent with what the RCC was offering them spiritually, emotionally, and financially, or to how they were being responded to. Any initial sense of their abuse being validated and the harms and the effects thereof that they had experienced being fully redressed, slowly transformed into an anxiety-producing awareness that they were caught up in a mere legalistic and often acrimonious treatment of their complaints and even of themselves as people.

At the end of each interview in de Weger (2020) all participants were asked whether their overall lives had improved or not after the reporting experience. All nine participants (along with many in de Weger 2016) felt deeply harmed by their reporting and/or disclosure experience. One could say that the survivors in de Weger (2020) were psycho-spiritually torn apart because their hoped-for healing based on frontstage promises thereof were traumatically replaced with expressions of other less victim-supporting though backstage beliefs, attitudes, and behaviours, which it appears were the more commonly held ones by officials dealing with CSMAA. The shock and subsequent confusion and harm experienced by such betrayal was for most even greater than that which they experienced as a result of the abuse (see also Table 1 in Pooler and Barros-Lane 2022, p. 6). As analysis progressed, it became clear that with almost all respondents, the retractions of compassion and belief appeared to be related to an inability or, indeed, unwillingness to genuinely and fully agree that what survivors were reporting was/is actually a serious issue of professional abuse enabled by heavily unbalanced power contexts. Now their whole Church was betraying them. Most simply could not fathom why and even how this had occurred.

What this present article proposes is that what participants were experiencing was being neutralised. How consciously devious or unconsciously cultural this neutralising

was on behalf of RCC officials, lawyers, and clergy involved is unknown. Either way, the impact was the same.

### 2.1. Neutralisation: Harming the Harmed

According to Anson Shupe's theory of clergy malfeasance and elite institutional neutralisation, when secret deviancies and corruptions occurring within such elite institutions, including the RCC, are exposed, the automatic reaction is to try to cover up these deviances. Shupe (1995) outlines the way elite institutions achieve this end through normative, remunerative, and coercive neutralisation techniques. These are employed to manipulate those threatening to expose such deviancies, be they victims, whistle-blowers, the media, or their supporters (Shupe 1995, pp. 86–95)[7].

Shupe's theory rests firmly on the foundation that clergy malfeasance and institutional responses thereto are about conflict and power, or intersections between the more powerful, and the less powerful and powerless (Shupe 1995, pp. 25–42; 2007, p. 56; 2008, p. 26). Utilising their power, the overall objective is to respond to the exposure of malfeasance in a way that does not publicly expose the institution's sins or/and threaten the institution's power, prestige, and wealth. In the case of CSMAA in the RCC, there is a further need to maintain secrecy regarding further realities concerning the lack of celibacy/chastity adherence amongst clergy in general (Sipe 2008). To put it more bluntly, neutralisation tactics exist to ensure the institution survives such exposure as cheaply and cleanly or undamaged, and as secretly as possible.

As a result of the interviews with survivor participants, while based firmly on Shupe's original list, a broader set of tactics and modes was generated to more fully encompass and describe the experiences of the participants in this study (de Weger 2020). More tactics and modes were required also to account for tactics found within other studies such as Kennedy (2009) and Benyei (1998). To this end, the following adaptations and additions to Shupe's original tactics and modes of neutralisation, particularly in the more refined context of the RCC and its responses to CSMAA, were developed (see Figure 1).

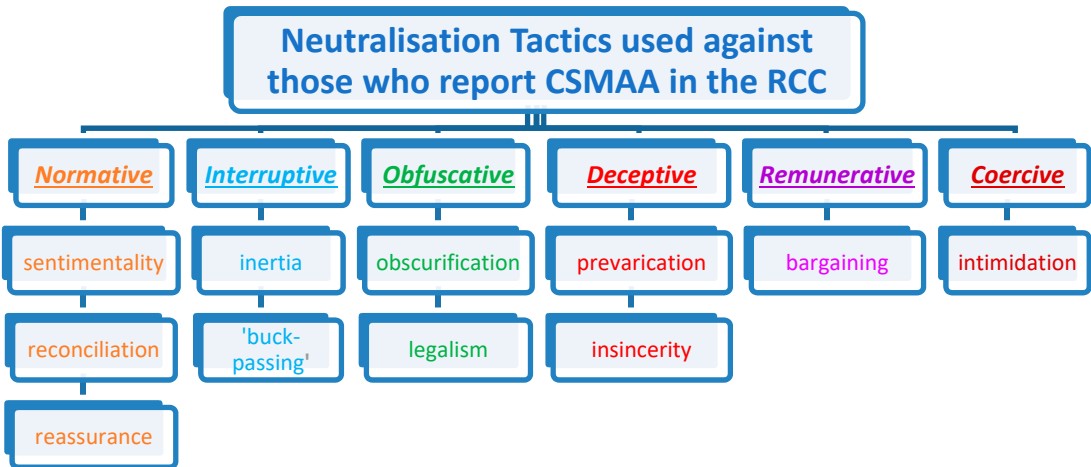

**Figure 1.** A new version of tactics and modes of institutional neutralisation (de Weger 2020, p. 229).

While the above figure presents a type of continuum, all these tactics and modes can occur at any time within the process.

The findings of de Weger (2020) were that neutralisation tactics, whether consciously or unconsciously employed, were clearly and frequently evident within the reporting experiences of the participants. These tactics and modes of neutralisation are a valuable skeleton around which one can flesh out actual events and experiences; they are a language which one can utilise in discourses of this otherwise neglected and even obfuscated issue.

### 2.1.1. The Normative Tactic: Modes—Sentimentality, Reconciliation, and Reassurance

The normative mode is that which seeks to not disturb the status quo. It engages a traditional 'Catholic' approach of deflecting possibilities of scandals within faith communities through a sentimental requesting of reconciliation and forgiveness. It also includes the pacifying approach of reassuring believers that things are being taken care of and to have faith in the goodness and authority of the church's leaders.

Sentimentality

At the time of Adela's reporting (first half of the 1990s), she experienced many requests to "calm down" and even to "stop victimising" her abuser. However, these requests did not only or directly come from the clergy or RCC officials. Adela tried to also convey that at that time, many lay Catholics could not deal with the inner conflicts clergy sexual abuse generated; their faith in their institution was being severely challenged. Accordingly, they just tried to stop people 'rocking the boat' and used sentimentality to do so.

Reassurance, and Reconciliation

The modes of reassurance and reconciliation were the second most referred to in this study. Regarding reconciliation, the concept of forgiving, of not throwing the proverbial first stone at sinners, of compassion, are all very central to Christian/Catholic teaching and spirituality. There is much tension between such central teachings and others such as justice for the oppressed, the injured, and the less powerful. It is into this tension that reconciliation can be used to manipulate victims to forgive their perpetrator and as such, to not further pursue complaints about CSMAA that may reveal clergy sexual activity to the wider public. As Adela's experience above shows, reassurance and reconciliation as tactics are closely related with sentimentality in that sentimentality is often the catalyst used to generate emotions needed to facilitate reassurance and reconciliation. The tactic seeks to reassure victims that officials are taking care of matters (even if they are not) and therefore, they, from their side, need to become more like Christ, more Christian by forgiving and forgetting. The insinuation is that if they do not forgive, they are not being good Catholics. In Adela's case, Adela said [Fr D] was quickly removed from ministry, which had the effect on her to perhaps become less combative with [Religious Order A]. However, because the action taken by [Religious Order A] was short lived she became less reconciliatory, and more vocal. Adela had found out that her abuser was back in [City A] to give a talk. Adela wrote to the then provincial and said

> if he comes to [City A], we are going to be outside picketing saying, you know, this guy is a sexual abuse perpetrator. And we will be picketing outside. So, I'd like to warn you now, to warn him off. And he's not to come to [City A] anymore.

> The talk was cancelled.

All usually working together, sentimentality, reassurance, and reconciliation are, Shupe says, "normative" tactics. Their primary purpose, whether intentional or unconscious, is to defuse victims' and whistle-blowers' anger and sense of injustice. This is done by appealing to their sensitivities and Catholic loyalties, to not pursue complaints any further because doing so will harm their Church and community and possibly them as well. As such, when successful, abuse events and their perpetrators are reabsorbed back into the institution's life and culture, and normalised even more, which was the desired outcome all along. Furthermore, as Shupe explains, this then has the added and disturbing effect of an increase in secondary deviance. Secondary deviance is behaviour that has had witnesses and which because of inaction, becomes more seriously entrenched into an institution's (secret) culture (Shupe 1995, pp. 46–48).

### 2.1.2. The Interruptive Tactic: Modes—Inertia and 'Buck-Passing'

The interruptive tactic is one which seeks to stop, constantly slow down, or interfere with the process of fully resolving reported CSMAA. The end hope is that those reporting

will become frustrated and give up. It is the tactic that is first employed for both individual reporting and often for such as media exposés. One of the first ways to achieve this is to remain inert.

Inertia

Inertia is the preferred first response as it requires the least amount of energy and financial input from the RCC. By doing nothing, those dealing with survivors reporting their abuse hope that the complainant will cease and desist. It is interruptive in that it seeks to disrupt or suspend, through the passing of long blocks of time, CSMAA cases that have been reported.

Vince was a young refugee entering the priesthood but was not yet old enough to do so. He was placed under the care of a priest from his own home country while waiting to enter the seminary. That priest groomed and coerced Vince into having sex with him under the guise of helping him overcome sexual temptations with women. His case, when he later reported it, showed many elements of inertia from the start even when simply trying to contact RCC officials or to get them to return his calls. Inertia became more obvious in the waiting games that he experienced on behalf particularly of [Bishop A] and [Archbishop B] with whom he had to deal. In the end, Vince had to get the help of an insider, "a friend who is a solicitor who had a brother, who was a parish priest" to try "to whisper in the Archbishop's ear". This was the first of many go-slow phases.

Inertia is easy for large powerful institutions but very painful and often traumatic for powerless victims. Brian, spiritually and sexually manipulated and abused as a young 18-year-old postulant in a Religious Congregation, described inertia before, during, and after his reporting process. In fact, this slowness of response caused him so much grief to the point where he was considering a new inertia-based complaint. This time the complaint would be about the Towards Healing lawyer, because of his very slow and then aggressive responses and how these had harmed him even further.

'Buck-Passing'

The second mode in the interruptive tactic is 'buck-passing'. Shupe did not include 'buck-passing' in his theory; however, he alludes to it as "bury[ing] a complaint in a maze of red tape, [and] bureaucratic confusion" (Shupe 1995, p. 83). For Kennedy (2009), however, 'buck-passing' was recognised as a commonly used tactic. She defines it as individuals denying responsibility, and an inability or unwillingness to fully follow any sort of process (Kennedy 2009, pp. 177–78). In this present study, many examples of 'buck-passing' could be found. One of the worst cases was Vince's. At first, Vince approached his Archbishop:

> I wrote to [Archbishop] at the, um, the Archdiocese office because I didn't know about Towards Healing back then. [Archbishop B] passed it on to [Bishop A], who replied that he thought the Church doesn't have the expertise and the authority to investigate these matters and that he needed to go to the police instead.

At first Vince did not want to go to the police. He felt very uncomfortable about doing so and as the abuse had started in a different jurisdiction Vince felt it would be too complicated, and he could not cope with that. After feeling dismissed by the Towards Healing officials, Vince did decide to report the abuse to the police. However, the police did not interview him but just took his written statement. Nothing came of it, as he said.

It did not end there, however. What Vince was told about the Church not having the expertise and the authority to investigate adult abuse was also not true as Towards Healing had dealt with adult cases before. His case was eventually taken up by Towards Healing, but it did not end well at all. Vince's case, discussed further in other neutralisation contexts below, has also gone on for almost a decade; it is still unresolved and ongoing. Furthermore, the bishop who dismissed Vince at the start was later removed from the Towards Healing case due to his every obfuscative responses to Vince, and to victims in other cases.

Whistle-blower worker in Catholic Education, Rhonda, provides another example of 'buck-passing' to neutralise both herself, and the complaints of a teacher, Julia whom

she was supporting. Julia had been groomed and manipulated into sexual activity with a Religious Brother, the director of the Catholic Education department. He had told Rhonda that he had "fallen madly in love" with Julia and did not seem to understand why this was wrong. Rhonda tried to explain why but it did not seem to make any difference. She started supporting Julia who soon began to realise that she had indeed been groomed and abused by a man who was both her boss and a Religious Brother and she wanted it to stop and to be reported. She had by now also heard, through Rhonda, of other cases against this Brother. However, both she and Rhonda soon began to experience the full frustration of institutional 'buck-passing' and inertia. Rhonda explains:

> I approached the Towards Healing manager when he was in [City] and said, "why haven't you done anything?" and he said, "It was an industrial matter". And I said, "It certainly wasn't an industrial matter". And he said, "Well, we're not touching it". And so, every time, [Julia] had to initiate everything. They'd never ring [Julia] back. Or when they did answer they'd say, "I'll get back to you in five minutes", but they never did. And [Julia] just wasn't emotionally equipped to do it. It was too hard.

Thus, in the case of at least one of the victims of [Brother], 'buck-passing' worked; Julia was neutralised, and so was Rhonda for the time being at least. There are examples of other methods of neutralisation here, such as prevarication and intimidation, all working to silence and cover up the abuses of [Brother]. [Brother]'s abuses have never been revealed to the public and he has been moved on to become principal of another [Religious Order] college in an Asian country.

### 2.1.3. The Obfuscative Tactic: Modes—Obscurification and Legalism

The obfuscative tactic employs both confusion and omissions so as to allow RCC responses to avoid responsibility for acts of CSMAA and even crimes. Alternatively, complainants are bamboozled by complex legalistic terminology. The obfuscative tactic also may employ passive neutralisation through definitional ambivalence (de Weger and Death 2017, pp. 228–29). When victims reporting CSMAA are confronted with obfuscatory modes, they are left dis-empowered and without any framework around which to establish claims of abuse. In such complexity and/or vagueness victims of CSMAA who are often already reeling from the effects thereof, simply cannot cope, and give up.

Obscurification

Obscurification is the mode which seeks to muddy the waters regarding perceptions of clergy sexual misconduct and even crimes and to conflate these with lesser concepts such as consensual affairs, as the proverbial Eve tempting Adam, clergy sin or weakness—anything but actual and much more serious professional misconduct resulting from grooming and manipulation of adults who may be in a vulnerable state. Obscurification, therefore, includes the lack of a clear, informed, professional definition of CSMAA. Such a lack has led to minimisation, denial of responsibility, and/or re-abuse. Obscurification is also a more subtle mode, which is perhaps why it has worked well to date. While there does not exist a clear informed definition of CSMAA, the perception that "probably it was the adult who was the instigator", and worse, will continue (Royal Commission 2015, p. 16048; see also Doyle et al. 2006, p. 59; Kennedy 2009, pp. 46–47, 127, 139–42; Byrne 2010, p. 44).

The lack of a universal definition of CSMAA needs to be analysed much more. To date, by definition, the only adults who could be abused are those who fell within the narrow definitions of "vulnerable adult" or people with mental or physical disabilities to the level of rendering them "like children" (The Vatican 2019a, 2019b). In the most recent drafts of RCC policy in Australia, "vulnerable adults" has given way to "adults at risk" and "adults with diminished capacity" (Australian Catholic Safeguarding Ltd. 2021). While the term "adult" is used when talking generally about clergy misconduct, when the discussion is more specifically about abuse, the document almost always uses the terms "adults at risk"

and "adults with diminished capacity" even though the definition for "adult" provided is as follows:

> Adult: When used throughout the NCSS document this is an inclusive term referring to all adults, including adults at risk (Australian Catholic Safeguarding Ltd. 2021, p. 22).

As such, what type of adult can be definitionally abused in Australia is still obscure. This also appears to be the case even with the *New Book VI of Canon Law* (The Vatican 2021), which was sensationally headlined as being a radical change regarding CSMAA[8]. However, when pushed for further clarification as to the type of adult that according to Canon Law definitions can be abused, even with this clarification, the matter is still very obscure (see Note 6). Ironically, perhaps, clear definitions of professional sexual misconduct against adults have been achieved by almost all secular professions. However, for the RCC, there is still no universal and canonically relevant definition thereof. In this vacuum, it is too easy for many to simply fall back on outdated perceptions of more or less harmless affairs between consenting adults.

This was indeed the dominant perception that all of the participants in this study experienced in some form. However, such beliefs only became evident when officials' true feelings were revealed, sometimes accidentally and often in anger, because victims were not passively submitting to the official processes. Four of the nine participants mentioned how this definition or 'belief' that they were not 'vulnerable' adults and therefore, what they experienced was not CSMAA, but more likely consensual sexual activity, albeit with an RCC celibate. For three of those four survivors, this was directly relayed to them as the following comments show[9]:

> He is quoted as having said he was not prepared to negotiate because "unlike child abuse, this was an adult thing" . . . I believe that everyone except for [Facilitator B], believed that it had to be consensual because I was over 18, and that's what the law says. And I believe that's why they fought it so hard (Lynne).

> On a couple of occasions when the abuse came up this priest still maintained [years later] that he was not entirely sure what constituted abuse when it came to adults (Elizabeth).

> Basically, what they were acknowledging is that, um, they pin it down to a consensual relationship. So, because you know, because it happened after I was 18, they consider it was a consensual relationship in me and him. So that just got them off every hook, you know (Vince).

The remaining five survivors expressed similar understandings that this was also how they felt they were viewed.

The other common ploy, somewhat related to the you-were-an-adult perception, was to suggest, sometimes quite directly, that victims were morally weak thus causing them to engage sexually with the cleric:

> [The psychologist paid for by the Religious Order] had been generally challenging, rather than supportive, but in the fourth session she actually suggested I might be feeling guilty because it was appropriate that I feel guilty! I was an adult after all and [Fr D] hadn't used force (Adela).

> And it was you know, the woman reeling him in. But then again, [Fr P]'s whole thing was, you know, it's your fault, or you know that you've sinned. It, in a sense, you've been complicit (Clair).

> [The psychologist from Centrecare] said: "Well it was just an affair between a boy and a girl, the boy left the girl, and the boy was a priest!" [Fr I] said: "You had an affair with him and now you ask us to pay?" (Sue).

The new/current Executive Director of Catholic Education Service stated one of the victims was a "prostitute" (yet had never met her). Of course, she wasn't (Rhonda).

None of these perceptions show any depth of knowledge of the dynamics of power imbalance in such contexts even though almost all other professions now clearly assert its existence. Nor do they appear to even want to acknowledge the possibility that the clergy in question could actually be a perpetrator of sexual abuse. However, as Kennedy (2009, p. 131) asserted, CSMAA occurs not because there is a vulnerable or even weak adult but because there is a powerful clergyperson willing to take advantage of their power.

As long as there is no clear universal informed definition of CSMAA, one placed squarely within a professional framework, myths and stereotypical beliefs will continue to determine how CSMAA is interpreted and how victim complaints, their personalities, and their needs are responded to when they report. As shown above, the responses to the participants in this study at least almost invariably centred on one major aspect—that they were an adult and therefore, the sex must have been consensual. The outcomes of complaints also reflected this. While some expressed consent confusion, it can be firmly stated that objectively speaking, none of the nine participants here, or the twenty-nine respondents in de Weger (2016), expressed any true and mature consent: all were groomed and manipulated into sex with their clergy abusers. Definitions of CSMAA must reflect this.

### The Role of the Cultural Secrecy in the Lack of Definitional Clarity

Why has the RCC not presented a uniform, un-obscure, and psychologically/harms-based definition of CSMAA? Part of the reason at least is that the RCC fears clarity even about what celibacy and chastity themselves mean (Sipe 1995, pp. 55–56; Bordisso 2011). In Canon Law, there obviously exists a clear mandate that "Clergy are obliged to observe 'perfect and perpetual continence,' (c. 277, §1)" (Provost 1992, p. 630). However, it appears that for 50% of clergy this definition is no longer the accepted one and are defining celibacy, chastity, and sexual activity for themselves (Sipe 1995, p. 61; 2016; Bordisso 2011, pp. 31–59, 60–66). For example: in the documentary *Secrets of the Vatican* (Thomas 2014) the following was included:

> NARRATOR: Francesco, a former seminarian, says he had a relationship with a priest who has since risen to a high position in the Vatican.

> FRANCESCO CACACCE: [through interpreter] He kept telling me that he was giving me his body, but his soul belonged to the Church, to God. The reasoning goes, "I'm a priest, but I have this need. I'll satisfy it and then go back to being a priest." It's a bit like vestments. "I wear them, I'm a priest. I take them off, and I'm just like anyone else" (Thomas 2014, 46 min 27 s).

As such, when it comes to clergy acting out sexually, this non-clarity needs to first be faced. Without such clarity, it is almost impossible for adults to construct or make claims that they have been offended against. The fact that definitional reluctance exists lies at the basis of presenting *obscurification* as a mode of neutralisation. With a clear universal, legal, and professional-based definition of CSMAA, the RCC cannot dismiss claims of CSMAA as merely consensual sexual activity albeit with a professional celibate employee of the RCC, or as merely clergy sexually misbehaving.

### Legalism

The other mode of obfuscation is legalism. Legalism refers to when legalistic approaches usurp and override the expected pastoral ones, i.e., legalism is used to confuse, oppress, or even frighten the person reporting CSMAA. Legalism may not at first appear to be a mode of neutralisation. A legalistic approach can simply be the application of a legal framework to processes of complaints of CSMAA. However, legal-based threats can and did appear to have been used also as attempts to close down complaint processes. Infusing legal jargon and the sense of lawyer superiority into the process created an atmosphere

where victims felt overwhelmed and rendered vulnerable because they were unable to understand or process such legalisms. While this may not be the intent, perceptions of legalistic threats succeeded in getting at least half of the survivors in this study to submit to the RCC process; it was either that or withdraw from it altogether. Of all the neutralisation processes, it is this one which demonstrates so well, what feminist standpoint theory scholars, Wylie (2003) and Jecker (2007), tried to explain as being "top down" and therefore, inadequate standpoints by which to understand the realities and needs of those who were in the "bloodied dust" (West 1998)[10]. Furthermore, the suggestion is always that those in such high places have little perception that they are like this. As such, legalism and legalists involved in the Towards Healing process could be seen as participating in a neutralisation process. Whether purposeful or unwitting, it has the same effect, and is simply not pastoral. Survivors in this study and in such as Kennedy (2009), were often faced with bamboozling legalistic processes, which firstly, they simply were not prepared for, and secondly, tended to "pull the rug from under one's feet" when it came to having some semblance of personal empowerment and control when approaching Towards Healing meetings and the like. The reigns of control quickly became firmly placed into the hands of the legal and financial representatives.

In Vince's case, he requested a review after things went very wrong. He said that included in that document was an acknowledgement that Towards Healing was meant to be a pastoral and not legal process; however, Vince found the document as a whole very difficult to fully comprehend and very "off putting". It is little wonder that he found this to be the case given that it was drawn up by a solicitor, who was also the CEO of Towards Healing in [City F], and a document which was highly legalistic in tone and language[11].

In such states of confusion and surprise brought on by the sudden and unexpected more legalistic approach, many participants felt tricked as it were into submitting to thinking that this was part of the pastoral process and thus submitted to that process. Those who expressed shock, dismay, and anger when realising that their pastoral hopes were somehow being replaced by legalistic and financial bargaining, soon experienced either a more threatening or a more dismissive approach towards them. However, others submitted, though reluctantly to this new confusing legalistic process. The end result of that process favoured the RCC in that such an approach was also successful in keeping the reality of the CSMAA within the RCC's files and under their control.

Lynne, someone who has now processed everything she has been through, was very helpful in describing the "bamboozling maze" that faces those who report clergy sexual abuse, especially CSMAA.

> One needs to distinguish between the National Committee for Professional Standards office (NCPS) and the Church rep., say Bishop or Provincial Leader (in my case) who are part of the process. But the NCPS representatives are paid by Catholic Church Insurances (CCI), so they are compromised too. My understanding is, despite that they're supposed to instruct the solicitors and barristers, I would say it was the other way around. And so, I would say that [Lawyers' Company] were running the show, really.

> Their job is to be instructed by the client [Religious Order E]. But I don't believe that the clients really understood what they were doing with me. I think they left them to it. [Lawyer's Company] probably ran things by them and said, "I think we need to stall this more. I think we need to deflect this more. I think we need to see how serious she really is. I don't think this case should go ahead because she signed a deed of release. And, and we've got to try and stop it there". There are all sorts of reasons why they legally could spin it out for as long as possible. And [Lynne's lawyer] will tell you that he believes they spanned it out as long as possible, so that the money would add up and add up and add up and eventually, I would not be able to afford to keep going.

Whistle-blower Rhonda supporting victim Judith, had a similar account of her experiences with the legalism of RCC processes when she took out a civil action as a result of being pushed out of her career because of speaking out against the Director/Brother of her workplace:

> They used every piece of legal action, so every time I went to fight it, they'd change what they could do, you know, and so I'd have to go back and refer because it confused the hell out of me, and they use their lawyers at all times.

It needs to be remembered again, that in both Lynne and Rhonda (Judith's) cases the initial RCC processes found that the complaints against the misconducting clergy were accepted as genuine.

Lynne's and Rhonda's insights contain understandings which came well after they had been through most of the process. Thus, for most victims/survivors who have just begun approaching the RCC, especially those approaching with deep and often naïve hope and trust that they will be heard and embraced, are, without these deeper understandings, very vulnerable to being neutralised by this legalism. However, the dangers of such happening are, as with abuse itself, only possible if the more powerful clergy and/or RCC representatives are willing to abuse their power to take unjust, uncompassionate advantage of these women and men; to neutralise those who trustfully submit to them for help and understanding.

2.1.4. The Deceptive Tactic: Modes—Prevarication and Insincerity

The deceptive tactic is exactly as the term suggests. This tactic is employed to consciously take control of cases through lying, or absences of the whole truth. It uses modes which promise action but never deliver. In the case of this study, it presents frontstage claims of care and concern, but which eventually collapse into revelations of more actually-held negative backstage beliefs about the person reporting CSMAA and of CSMAA itself. Disingenuousness is the flavour of this tactic. For some, it is enough to crush them once discovered, to neutralise them, causing them to desist in their complaint.

Prevarication

The mode of prevarication is taken from Kennedy (2009, p. 178). Again, while Shupe alludes to this tactic, it is not included in his given tactics and modes. However, it is one which does require a place in the methods of neutralisation used within the narratives of the participants in this study and elsewhere. It is classified here as a deceptive tactic because as Kennedy (2009, p. 178) describes it, it is

> another pernicious ploy [used] to "string" women along; giving the impression that action was being taken but ultimately no progress or outcome resulted.

This mode really is a cruel and an even trauma-producing shock when it is discovered by victims reporting CSMAA. For the participants in this study, prevarication became most apparent in regard to requests that the perpetrator cleric be removed from ministry or dealt with in some way. For all participants, this was a major request included in their reporting process. For both Lynne and Adela, it was of particular importance. However, their requests, while acknowledged at first, had to be fought for constantly and over long periods of time, because while changes were promised, most of these ended being only minor.

> It had only been a matter of months and there was already talk of [Fr D] returning to full-time ministry. We protested but were assured by a church authority this was not so. Two weeks later [Fr D] was back in his parish (Adela).

> I asked the Provincial, [Fr I] about the Australian Catholic Directory and he said that he forgot to inform the publisher that [Fr H] left the order in 1999. However, looking back, I believed what the Provincial told me and didn't question more. As well, he told me [Fr H] left [City D] for [City E] because he received a scholarship

to study Asian Studies at [Name of University] in 1988. But after the mediation, a PhD student friend—[Name of Student]—at [Name of University] helped me to find out that there was no such scholarship awarded to [Fr H] (Sue).

[Facilitator B] said "[Fr K] has stood [Fr J] down from all public ministry and [Fr J] has volunteered to step down as leader at [Name of Church]" (I inform [Facilitator B] that [Fr J] is still listed as [Religious Order E]'s Provincial Vicar and Consultor). I told [Facilitator B], that [Fr J]'s name has been removed (on some websites), however, he has been left as Provincial Consultor & Provincial Vicar. I heard [Fr K]'s argument on this, that this is an internal not public position. I told [Facilitator B] that I found it quite incongruous actually, not to mention personally insulting, that he/they think [Fr K] can be left as a significant leader in an organisation that has otherwise stood him down publicly. This does not seem right or satisfactory at all to me. And two days before Christmas, two evenings before Christmas [a year later], I received formal notification by [Facilitator B] that [Fr J] was going to be fully reinstated. I was outraged (Lynne).

When, after promises of action, perpetrators are not dealt with justly, this can easily be interpreted as a deceptive neutralisation tactic: attempts are made to quietly reinstate clerics even after they have been acknowledged to be misconducting clergy. It appears that the expectation of those in power is that victims will either not notice or not respond. However, the harm such actions produce when they are almost inevitably discovered leave a sense that all the affectations of concern and desire to heal were merely empty prevarications.

The other main form of prevarication is revealed in promised apologies of meaning and substance never eventuating. Lynne explained how the lack of receiving a promised apology was a constant source of concern and pain. She described how she had written to [Facilitator B] saying that there had been no mention of a written apology in the settlement document and that she finally had to insist on an apology as part of the settlement. She then related how she told a friend that if she did not get a signed admission and apology then she probably would not be signing any papers. An apology in this case did eventually come, but again, not through any proactive desire on behalf of Towards Healing and [Religious Order E] to see Lynne healed.

Brian had asked for what he genuinely expected would be a meaningful apology. A form of apology did eventually come. However, he was shocked by its generic content:

The "apology" [was] weak and insufficient to me. The "harm done" was not specific. Very legally vague! I could have fallen and injured my toe at church one Sunday or been bullied at school due to my association with the [Religious Order F]. Sexual Assault or Sexual Abuse causing deep and chronic mental damage was not mentioned at all.

To add insult to injury, Brian also experienced prevarication in the form of offers to him to help organise a healing liturgy for victims, and to talk to seminarians on the effects of abuse. He was quite happy to do so, however, after the Towards Healing process was over, he has never received any further responses from [Bishop E] and his diocesan office concerning these. Brian had been promised much but received very little in the end.

In one sense, regarding those in this study, the whole Towards Healing process could be seen as organised and systemic prevarication. What Towards Healing promised in policy documents *Towards Healing* (National Committee for Professional Standards 2010b) and *Integrity in Ministry* (National Committee for Professional Standards 2010a), was simply not being delivered. Prevarication is a powerful but cruel reality in the reporting process.

Insincerity

Insincerity, while obviously part and parcel of prevarication, was a more omnipresent characteristic: it was found in all cases in this study. It was when this overall insincerity was realised by the victims, that new and even deeper trauma was inflicted. At first, there was an unwillingness or inability for most participants to accept the reality that the RCC might

actually be insincere in their responses to their genuine needs. In fairness to at least three of official first responders who were working at the more basic street level, this insincerity also appeared to have a hierarchy in that first responders were more genuine or sincere in their concern for the victim. However, as one went up the ladder of officialdom within RCC processing of complaints of CSMAA, especially up to the ones who made the decisions on legal matters and financial redress, the more apparent the insincerity became. As such, it needs special attention and much closer scrutiny as a deceptive mode of neutralisation. Insincerity is not as obvious a neutralisation tactic as, for example, bargaining. It is perhaps more a general though hidden attitude to those who report. However, as shall be shown, insincerity can have the same neutralising effects in the process of reporting CSMAA to the RCC, once it is discovered by the person reporting, to exist.

The insincerity revealed in this study may well just be a product of confused beliefs, attitudes, and behaviours which still need reforming through deeper analysis of CSMAA and professional sexual misconduct in general. It is not unexpected that in this vacuum of conviction that CSMAA is indeed abuse, that insincerity cannot help but become an underlying attitude in regard to the reporting structures within the RCC. This was certainly the case according to the nine participants of this study, at least.

According to Sipe (1995) and Doyle et al. (2006), insincerity regarding CSMAA is deeply rooted in RCC systemic, cultural, and social beliefs, attitudes, and behaviours. Furthermore, centuries of misogyny have embedded a deep suspicion of at least women and, therefore, female victims anyway. However, Shupe explains how these can change within a strongly centralised hierarchy of power as is the RCC (Shupe 1995, pp. 27–31). However, before any such changes can occur, a belief that such change is needed, and then a willingness to facilitate those changes, is still required.

Insincerity is founded on actual though more backstage beliefs and understandings of the nature of CSMAA. These underpin a consequential belief that those who make complaints of CSMAA are in error. However, the frontstage of the RCC obviously must still present a caring attitude to all clergy sexual abuse victims—hence, insincerity. The term 'insincerity' itself, is a rather mild or bland one; however, it is a central player in the production of deep harm. It is also expressed in many different forms. As such, it will be revisited in more detail below.

2.1.5. The Remunerative Tactic: Mode—Bargaining

Pre-Official Professional Processes

The bargaining process, according to Shupe is one which "*appeals* to not press for redress [through litigation] in return for a [smaller] financial settlement" (Shupe 1995, p. 86; 2007, pp. 76–78). Bargaining often begins when it has become clear that survivors are not going to be perfect victims. Survivors become imperfect victims when they do not succumb to the normative modes of neutralisation presented above. However, forms of bargaining can be found within those modes as well. For most, though, bargaining is a specific mode that usually includes offers of money and organised assistance, all controlled by the institution. In Australia, since 2000, the RCC's response processes of Towards Healing and the Melbourne Response have become a form of what could be deemed official pastoral bargaining.

Those who made complaints about CSMAA before these processes were established (Gary and Adela) tell a somewhat different story than those who reported their abuse later to such as Towards Healing. Perhaps because clergy sexual abuse had not yet become a major newsworthy item, there was less concern, as it were, in the RCC of people revealing CSMAA. As such, institutionally controlled bargaining was not as necessary. In Gary's case, he also believed he had "no case" to bargain with anyway, and should he have wished to take things further, he would have just been "fobbed off". For Adela, any financial redress was not really an issue because for her; she just wanted her abuse validated and for the priest to apologise. However, even more so, because she had by then discovered the existence of at least twelve other victims, she wanted her abuser priest to be removed from

the possibility of harming anyone else. Thus, at this earlier stage (1970s), there was no real pressure yet for a pastoral bargaining process in the minds of RCC officials. It was not far away though.

Post-Official Professional Processes

Regarding those who reported to the now-established Towards Healing process, a whole new set of attitudes and behaviours were presented by the RCC. The clergy abuse scandal had by this time been greatly reported in the media. Headlines, documentaries and dramatisations of clergy child abuse and church cover up were proliferating in the late 1990s and the RCC knew it had to do more to control the damage that the many cases that were coming forward could possibly wield. To respond to this growing pressure Towards Healing was created (Medley 2001, p. 82).

Towards Healing followed a very specific process and had its own Church officials/authorities, lawyers, insurance people, therapists and assorted assisting clergy, facilitators, and moderators. However, while it was promoted as a pastoral response it has become clear through studies such as Medley (2001) and Courtin (2015) that Towards Healing (and the Melbourne Response) appeared to have rapidly become a bargaining process concerned with private financial settlements obtained through the use of lawyers and very carefully worded legal documents, albeit under the banner of care and healing. Parkinson (2009), in his review of Towards Healing recognised the tension between the two elements of the pastoral and legal approaches:

> It follows from this that pastoral and legal approaches are not independent of one another, but they can be in tension. Resolving this tension between the pastoral and legal approaches is a critical issue—for both sets of demands are being placed every day upon the Towards Healing process, and are reflected in submissions to this review (Parkinson 2009, p. 10).

One argument of this study is that in regard to CSMAA in particular, because of a general underlying or backstage dismissal of such cases as not legitimate due to age, it appears that the legal approach tended to usurp the pastoral. This then results in the Towards Healing process taking on the strong bargaining tones presented in the participants' experiences, rather than genuine validation and healing of their abuse. This bargaining took various forms and included a range of added elements. While not all participants revealed clear bargaining elements, most did, and for some it was the very flavour of the Towards Healing process.

Sue, a young refugee, alone in Australia, was groomed and sexually abused by a Religious Order priest from her own country. When she approached Towards Healing, she had not intended to bargain. She wanted help and validation that what she had experienced was the same as what she had recently read about in the newspaper about other adult victims, and that she was not alone. She wanted someone to care about what she had experienced as the events still haunted her. However, instead, she experienced many painful and confusing responses by the RCC officials with whom she dealt. It became apparent from the start of the Towards Healing meeting, that [Fr H] whom she had deeply trusted, had taken on a very different demeanour towards her than he initially displayed, and started to minimise what had happened. As a result, her pain and anger put her on a new path. This new path was away from her previous naïve awe and respect for the RCC (she had converted to Catholicism after arriving in Australia) to one which then became a fight for her survival and happiness. In Sue's case, once she realised that the RCC was not really on her side, she stopped being a perfect victim. Depressed and angry, Sue then demanded what she believed was a more just redress. Doing so soon revealed more backstage beliefs about Sue's CSMAA; that according to [Fr H]'s superior, [Fr I] and supported by [Facilitator A], Sue had simply had "an affair" and now Sue just wanted the RCC "to pay for it". This devastated Sue.

Sue had not rolled over and just accepted the first bargains. She would no longer be the perfect victim, one who was naïve and submissive and even grateful for bargains made.

It was realised that she could not be easily dealt with. As such, it then becomes clear that, as with Adela above also, she, like many reporters of CSMAA and other forms of clergy malfeasance, become considered as "enemies". Sue's complicated, complex, messy, and demanding case/s left her feeling that she has never been genuinely heard and believed, and accordingly without true psychological, spiritual, or practical resolution. In the end, she accepted the presented offer or "bargain". However, now it was not happily or pastorally given but based rather on the belief that Sue had simply had an affair. Nevertheless, it became clear that she needed to be silenced by giving her the larger amount of money she requested. This larger amount, though still relatively small and used to pay for counselling and other related needs, was a little more realistic considering what she had been through and the effects it had had on her.

Of all the Shupian tactics that can be seen at play in Vince's case, bargaining would have to be the most overt. From the very start, Vince was offered $10,000 for his years of abuse in the hope that he would simply accept that and disappear, or not take matters further. When he did not disappear, he was offered $30,000. Both times, the money was "granted" as a way to "help him" in some way. At both times also, however, there was an undertow of, "while we don't really believe you deserve it, you really should take this":

> Well, it's already intimidating, all along because even at the facilitation; the two guys said to me, "now look, take it or leave it, because at the end of the day, it can be, it can be your word against the priest". And back at that time, you know, the, the, the statute of limitation was in place, and you got no choice, you got no chance (Vince).

This attitude was present even though Towards Healing had at the start acknowledged that Vince's was a valid and an obvious case of abuse. However, when it came to the bargaining or amounts of compensation, from what Vince disclosed, the RCC then suggested that the later sexual activity after he had turned eighteen, was not really abuse because he was "a consenting adult" and as such compensation should not be expected. According to an email Vince received by mistake, they also actually believed he was reporting CSMAA for his own gain only, with intentions to write a book about his and others' experiences of CSMAA. As such, the RCC official response was to neutralise him, using manipulation, falsehoods, intimidation, and with a final "take-it-or-leave-it" bargaining offer.

In one way, one could say that Lynne's case was also all about bargaining. As with Vince, Lynne's [Religious Order E] were not convinced of the gravity of [Fr J]'s clergy/professional sexual misconduct, believing instead that it must have been consensual as "she was an adult". As such, they believed the request she made for redress was outlandish and unwarranted. This resulted in a to and fro bargaining process which had at its base, the strongly-worded request by [Religious Order E]'s provincial that Lynne not pursue the matter further but to accept the financial settlement offer they presented. However, at that very point of time, Lynne had a series of surgeries about to occur and her mother was seriously ill as well. The [Religious Order E]/Towards Healing knew this.

> That sort of pressure was, for me with my history, like another [Religious Order E] priest abusing his power over me because he could. That is why I pulled out of the negotiations, accepted an offer I still consider an insult, and basically felt forced to call it quits from my end. The pressure, with only one week to go until hospital, was too much. I had to extricate myself.

Needless to say, Lynne found the whole process, veiled in threats and statements suggesting she take the money because there would be no other offers, was anything but pastoral. She had sought validation, healing, and for [Fr J] to be removed from ministry, and a level of financial reparation which would balance out all she had suffered and lost. Lynne felt intimidated and bullied throughout the process. In the end, she did not get the full validation she required, nor the financial support she needed, and [Fr J] was reinstated after 12 months. That was the last straw for Lynne: She decided then to begin a civil case against her abuser and the [Religious Order E]. This too ended in an agreed-to financial

settlement because of the protracted stress it was having on Lynne. The RCC had the bargaining power of endless legal processes and finances: Lynne of course had nothing like that.

For Brian, after having experienced the Towards Healing mediation meetings, it was his conclusion that these meetings were nothing more than bargaining sessions infused with pressure to conform (Gleeson 2015, pp. 326–29). These meetings were where the Towards Healing Lawyer [Lawyer B] told Brian that he had to "take it or leave it, by 4pm that day". By this stage, Brian had had enough. He took the gesture offered. For Brian, this was bargaining, pure and simple. The experience with Towards Healing and [Bishop D] left him feeling anything but validated and healed. His experience had come down to a bargaining process controlled by RCC employed lawyers and financiers as he clearly outlined in the written account he presented.

Bargaining takes many forms. However, for most it involves financial incentives. These incentives are also often presented in a take-it-or-leave-it manner. Such offers or bargains are also almost invariably given in a context of demanding no further action. As such, the underlying intent of preventing further, more costly and exposing actions on behalf of the survivor is very clear. According to Shupe (1995, pp. 92–93) elite institutions, including the RCC, when confronted with exposure of deviancy within

> follow a basic negotiation pattern that seeks the most containment of victim anger (through silence) for the least amount of money.

The content of the narratives presented here confirm this general approach.

### 2.1.6. The Coercive Tactic: Mode—Intimidation

The final and most extreme of Shupe's tactics/modes of institutional neutralisation is intimidation. Because it is the final mode or last course of action to stop full revelation and claims for damages, intimidation did not overtly appear in all participant narratives, but it was often present in covert expressionsof a general sense of intimidation. Those such as Elizabeth and Clair who did not make major demands or were comparatively happy with the reporting process even given their issues, did not make any references to overt intimidation. However, those that did make waves all felt intimidated to some extent, some quite severely.

Gary, whose CSMAA was the oldest in regard to time, did not point so much to any intimidation concerning his reporting of his CSMAA. However, as he became more of a whistle-blower about clergy abuse, Gary's willingness to speak up publicly became known to [Fr B], who the RCC had employed to manage the clergy abuse crisis and the media. As such, it was not so much what Gary was exposing, but who, such as [Fr B] and their desire to keep matters out of the press. Accordingly, Gary's character was publicly questioned by [Fr B]. Gary said he was "bad-mouthed" to the press by [Fr B], "on three separate occasions", where [Fr B] kept "saying that I had a hidden agenda". In one sense he did have an agenda, however that "hidden agenda", Gary explained, was that he had actually looked at commencing a civil action concerning his abuse as a child, because the police still had not gone ahead with any criminal charges. Again, this was another case of trying to silence victims to stop them exposing what the RCC knew were even credible and already acknowledged abuse cases. If victims of declared and always criminal child abuse by clergy can have such difficulty, there was even less chance at this time (1990s) of CSMAA victims such as Gary, being heard and validated, let alone redressed.

Intimidation, according to Shupe's theory, is more reserved for those who threaten civil or criminal action, and especially against those such as lawyers and media who support the victims in their civil or criminal cases (Shupe 1995, pp. 97–98). Such people are often then constructed and proclaimed by the Church as being anti-Church rather than people seeking justice. The victims themselves are often then also cast as suspicious or dubious in regard to their accounts of abuse, and sometimes even as unstable, or as false accusers. However, intimidation can also be revealed whenever a reporter or whistle-blower of deviancy becomes non-cooperative during the reporting process because of non-acceptance

of Church responses. The institution senses a threat and responds accordingly with this stronger tactic.

Adela experienced both direct and indirect intimidation but not just from RCC officials by making her feel like their 'enemy': She also experienced intimidation from other Catholics and broader circle of old friends within her community who decided to support her abuser after he had already presented to them his own version of what had 'really' happened. Adela was ostracised and experienced character assassination, not just by her abuser and his [Religious Order A] confreres, but by her friends and others connected to [Religious Order A]. She was accused as being the instigator who had "thrown herself" at "poor Father".

In Sue's case, once she requested a more realistic and just redress, the sense of intimidation began. She described how she admired this priest at first but at the Towards Healing meeting, he glared at her, and she actually became a little frightened of him. His whole demeanour towards her had changed. Furthermore, the statements the earlier Towards Healing-selected psychologist suggesting it was "just a boy-girl affair that went wrong", could well be seen as forms of character assassination and intimidation, or at best an attempt to manipulate Sue into feeling guilty, frightened, and unsure of herself. Covertly, what they were doing was "demanding" that she "cease" steps towards civil action which is the raison d'être of the coercion tactic (Shupe 2008, p. 75).

For some reason, perhaps due to his case being one of same-sex misconduct, Vince experienced intimidation from the start. Even though during the Towards Healing process, Vince had not mentioned he would seek lawyer assistance and take his case to court; he was intimidated in a way that suggested that the RCC suspected he might do so. Vince mentioned at least three specific times when he was made to feel that he simply "had no chance" of winning such a case in court. This is not an example of a pastoral, compassionate, and just response to someone who was obviously groomed then abused when he was very vulnerable, manipulated into a long-term dependent but sexually based service of his abuser who controlled the whole context. Instead, the RCC responded with threats, dismissal, cruelty, and lies in order to neutralise Vince and the possible exposures he might make revealing the RCC's inner secrets of clergy sexual activity and lax attitudes thereto.

One possible reason for Vince's appalling treatment is, perhaps, that he was also becoming someone who supported other victims of adult and child abuse from within his community. These activities were known by the church. He was not only a victim himself, but he was also becoming a whistle blower who, according to Vince, the RCC feared may write a book about all his experiences. As such, he had to be "shut down" . . . neutralised. However, he had not set out with any intentions other than simply wanting and believing he would receive understanding for his abuse and its consequences in his life. He had needed decades of therapy to help him deal with the harm that the abuse had caused. Instead, when asked to sum up his experience he had the following to say:

> For me and from what, from my experience it's got nothing to do with compassion and anything, but they just weighed it up on the ground of their legal advice and try to find a way through it so, in my particular case, I think they, they um, got the upper hand, they say, look, we can just pin it down to sex, um, a homosexual relationship, you know.

One of Vince's final comments was "So, it's continuous, it's intimidation all through". The painful official RCC responses included current Archbishops and Bishops, their own NCPS and RCC financial people and lawyers. This was not a pastoral experience but one which resulted in further deep harm (see also Table 1: Pooler and Barros-Lane 2022, p. 6).

While Lynne had felt intimidated during the Towards Healing process, it was in the court case that the RCC began using intimidation to the full:

> they certainly fought me as hard as they could possibly have fought on the ground that it was consensual, and I mean they pulled in QCs over this. You know, I was up against seven . . . seven, um, . . . top legal minds, by the end.

Such intimidation could be simply a part of Australia's adversarial legal approach. However, given that the [Religious Order E] and [Fr J] himself had acknowledged the CS-MAA, and given the Towards Healing document's descriptions of clergy sexual misconduct and power abuse (National Committee for Professional Standards 2010a, pp. 7–8), and then the use of intimidation and character assassinations during the civil action, it could be said that the case was not conducted "in good faith": "good faith" is an underlying requirement of the adversarial system (Gibbs Wright 2020). Lynne says that her lawyer protected her from the worst of the intimidation and character assassinations but still, a great deal got through.

One of Brian's last comments as a participant was that he "does not thank the RCC for arriving at this state", the state of confusion and brokenness resulting not only from the abuse but the reporting thereof as well. The need to make such a statement reveals anything but a pastoral attitude. It is a statement inculcated by an ever-present undertow of intimidation. That undertow made its presence known in one climactic event.

Brian had been trying to ascertain what it was that Towards Healing were offering. When Brian said that the church would have to pay his legal fees (on top of the money offered), the Towards Healing's lawyer slammed the mediation table with his fist exclaiming, "The catholic church doesn't have to do anything!" Such slips of the tongue, often expressed as anger rises, reveal the hidden true beliefs about victims of CSMAA even when there was/is clear evidence that the abuse did occur and that such men and women have indeed been traumatised by that abuse. Yet, they are at the same time seen as impudent, as money hungry, as seeking redress where they do not deserve it because they were adults at the time of the abuse. Most come first and foremost to seek understanding and comfort. Brian was still seeking this only two years ago when he went to visit another Bishop who did provide that comfort, having himself also been a victim of CSMAA before becoming a Bishop. One then needs to ask: Why does the RCC in this case, feel the need to approach mediation with victims in such negative, defensive, and even intimidating ways?

In her interview Rhonda described how the victims of [Br A] were being slowly and deliberately intimidated through harassment leading to stress and related poorer performance in their jobs. In turn, this led to dismissals or non-renewal of work contracts. In the case of whistle-blower Rhonda herself, intimidation came in a similar but somewhat less subtle, "if you want to keep your job, keep quiet". However, Rhonda could not keep quiet, which resulted in more intimidation. Rhonda explained how within the [City H] CEO there was a woman [Senior Staff Member Ms B].

> [Senior Staff Member Ms B] managed, for the bishop, these sorts of situations many times. She's a bit of an expert, and she immediately called the head of the [Religious Order I] who was featured highly in the Royal Commission.

> The [Religious Order I] Provincial arrived and asked people if they wanted to talk to him about what had been happening. A colleague of Rhonda who'd been at Cath. Ed. for 20 years, said to her:

> Don't you dare go near him. He's not trustworthy. He only comes here when there's problems. So, don't you open your mouth or go near him. Because he's not a trustworthy man. He's just been bought in to cover this up, right.

It is clear then that this workplace was one experienced with clergy sexual misconduct. Indeed, [City H] educational institutions figured highly in the Royal Commission into Institutional Responses to Child Abuse. As such, [Senior Staff Member Ms B] would have suspected that neutralisation modes such as sentimentality and reassurance/reconciliation would not work on such as Rhonda. Such people as Rhonda were already hardened because of their experiences but also because most people working within RCC offices are not usually naïve when it comes to clergy behaviours. However, those in high positions in such as Catholic Education also have enormous power to hire and fire, to destroy or manipulate. One of Rhonda's final comments expresses clearly how the intimidation mode operates:

> I met with [Bishop E] for three hours recently. He seemed more worried that there were still victims working at Catholic Education. He thought he had got rid of all of them. I suggested to him that because he and his senior executives treated several people who were abused or witnessed the events, so poorly, that his church's behaviour sent out a clear message to others to not talk/report.

This is the direct and indirect power of intimidation. It works particularly well when the intimidator knows how vulnerable and weak their victims are. It is also often successful. Those in the RCC seeking to maintain the cover-up of CSMAA, know this and this is why it is employed, or better still, this is why, as Lynne, Rhonda and others experienced, they employ lawyers and others to do the intimidation for them.

*2.2. Insincerity Revisited: The Unexpected Shock Wave*

If there was one term that most all-encompassingly defined the experiences of reporting in this study, it would have to be insincerity. Regarding those dealing with CSMAA victims, the belief that such people are probably just guilt-laden puritans trying to blame clergy for their own sin, still reigns, at least backstage; either that or they are people seeking to undermine, embarrass, financially exploit, and/or even abuse the Church and its representatives. Regarding the survivors' accounts in both de Weger (2016, 2020), there was no evidence for any of these erroneous held perceptions. One of the opening questions asked of all participants was, "What were you looking for when you approached the RCC". All said, healing, acknowledgement and validation; four needed financial help as a result of sequelae of harms; and most also wanted removal of the offender, not in vengeful rage but in order to protect others. All loved their Church. Ironically though, it was the RCC officials who believed that it was the victims reporting CSMAA who were being insincere. The evidence in this and other studies simply does not support this perception. Rather, evidence is much more supportive of the RCC responses being insincere.

Insincerity in regard to RCC responses to reporting did not have just one singular expression. Throughout the research many varied though related forms of insincerity revealed themselves clearly. Below is a list of these various expressions as well as how they most relate to other tactics and modes of neutralisation. Indeed, these various expressions need to be more singularly revealed and discussed in order to explain how they can function as particularly the deceptive tactic and modes of neutralisation. While all participants implied insincerity, the (pseudo)names of only those who provided more explicit examples of the discussed elements shall be given.

2.2.1. Biased Psychologists (Normative/Deceptive)

Church/Towards Healing psychologists often steered perceptions of the CSMAA towards RCC desired outcomes (Sue). On the other hand, victim-chosen therapists did not hesitate to describe and explain the CSMAA as clearly abusive (Sue; Lynne; Brian; Clair; Adela).

2.2.2. Lack of Action on Offending Clergy (Interruptive/Deceptive)

Offending clergy were not removed from ministry unless the survivor pushed for such; many were simply moved to other dioceses in Australia and overseas, some were even promoted even with the knowledge they had misconducted (Sue; Lynne; Clair; Vince; Brian; Gary; Adela; Rhonda).

2.2.3. Be the Perfect Victim or Else (Coercive)

Victims/survivors were poorly treated if they ceased being perfect as in fully compliant victims and even good Catholics for the RCC (Brian; Sue; Lynne; Vince; Rhonda).

2.2.4. Slips of the Tongue (Obfuscative/Deceptive)

The existence of dismissal of victims and even contempt thereof was revealed when officials became angry or felt challenged by the complainant. At these times, true feelings,

and stereotypical or RCC-normalised definitions of 'consensual affairs' or at best, clergy simply misbehaving or sinning, were revealed in both verbal and written 'slips of the tongue'. These more real 'backstage' beliefs, attitudes, and behaviours also became even more evident in cases of RCC official dealings with whistle-blowers of CSMAA (Brian; Sue; Lynne; Vince; Adela).

### 2.2.5. Support Person's Status Influence (Deceptive/Remunerative)

While it should not be the case given the stated mission of such as Towards Healing (National Committee for Professional Standards 2010a), successful outcomes of official processes were sometimes based on the presence of powerful or important support persons rather than the veracity and reality of the CSMAA, and the opposite is true: (More powerful/important support person—Adela; Lynne: less powerful/important support person—Sue; Vince).

### 2.2.6. Character Assassinations (Coercive)

Character Assassinations were often only backstage conversations between officials but sometimes became more overt when victims did not submit happily to official RCC processes and threatened to take further action. They became quite overt when RCC officials were dealing with CSMAA whistle-blowers (Sue; Lynne; Gary; Vince; Adela; Rhonda).

### 2.2.7. Lack of Sincere and Meaningful (Healing) Apologies (Deceptive/Obfuscative/Normative)

Most participants complained that the apologies they received were generic and lacked substance as to what had actually occurred, and any clear admissions of actual guilt (Lynne; Brian; Elizabeth; Vince; Adela; Rhonda).

### 2.2.8. Obfuscating or Confusing Legalism (Obfuscative/Deceptive/Coercive)

Towards Healing and newly developing RCC protection policies were/are presented as a pastoral process. However, as many or most participants experienced, a more legalistic approach came into play usually very early in the official complaint procedures, often confusing victims who were seeking merely a pastoral response (Clair; Brian; Lynne; Vince).

### 2.2.9. Lack of Adequate Redress (Remunerative/Coercive)

While two participants (Claire and Edith) were more or less happy with the financial redress they were given, along with the assistance provided for counselling, most in this study and de Weger (2016) felt their lives as a result of CSMAA were far from compensated. Furthermore, it was apparent that the RCC always began with the least amount of financial redress that they could (Sue; Vince; Lynne; Brian).

### 2.2.10. Secrecy (Deceptive)

There were clear and frequent examples of the RCC wanting to maintain CSMAA events as secret. This was achieved through the new official responses being essentially bargaining processes to maintain the silence of the victims through non-disclosure agreements requiring promises of no further action. Doing this ensured RCC kept full control over their stature, finances, and its many other secrets regarding clergy sexual activity. It was also achieved through sentimentalist pressure to keep quiet so as to not harm the RCC 'good name' (Sue; Lynne; Clair; Rhonda).

### 2.2.11. Abusing the Deep Power Imbalance, Again (Coercive/Remunerative)

This was especially the case for those who were more vulnerable at the time of self-reporting (Sue, Vince). Those who were naïve or uneducated regarding the inner realities of the RCC—or who did not have strong support people, fared worse or were taken advantage of more than those who were more educated about the CSMAA and/or who had high-

status support persons. However, those who may have been more educated were given similar treatment if they proceeded with civil actions. (Lynne; Clair; Adela; Brian; Rhonda).

### 2.2.12. Clericalist Elitism (Coercive/Normative)

This element was also evident in the way victims were somewhat initially seen as the cause of the CSMAA events, or, as people of questionable character trying to besmirch the good name of the church for personal or monetary gain (Gary; Adela; Sue; Vince; Brian; Rhonda). There was also evidence of sympathy rather than true empathy which maintained clergy self-perceptions of strong clergy—weak laity resulting in subtle transferences of guilt from clergy to victim. This clericalist elitism came through particularly when RCC officials were dealing with whistle-blowers as opposed to victims, or victims who exhibited whistle-blowing elements in their reporting (Gary, Vince, Rhonda).

### 2.2.13. Lack of Professionalism (Deceptive/Remunerative)

While the RCC through its National Professional Standards Office sought to bring the clergy under the umbrella of the term professional, there was often a concerning display of lack of professionalism. This was evident both in pastoralism camouflaged as legalism, and in the simple but often serious mishandling of victims' complaints, and indeed, lives. It was also evident in the advice given by officials as to how to manage Medicare issues relating to compensation (Vince, Brian, Clair).

### 2.2.14. Lack of Clear Definitions (Deceptive/Obfuscative/Normative/Interruptive)

RCC documents such as *Integrity in Ministry* (National Committee for Professional Standards 2010a) affirmed in words that CSMAA with any adults, not just the definitive "vulnerable" or "at risk" ones are a possibility, and one which called for professional responses through pastoral approaches. However, evidence from those who have actually experienced CSMAA and reported their experiences revealed that CSMAA was not being perceived thus—as misconduct perpetrated by professionalclergy—but rather as religious pastors falling, experimenting, sinning, or breaking the rules of celibacy (Sue; Vince; Lynne; Brian; Clair; Rhonda). Such perceptions and contexts carry with them, therefore, expectations of forgiveness and empathy (for the cleric), and spiritual family silence as opposed to legal consequences or public reprimands (Benyei 1998, pp. 104–17). With no clear and more reality-based definition of CSMAA, erroneous perceptions and their repercussions will only continue.

### 2.2.15. Summary of "Insincerity"

Most ordinary people would likely agree that people or institutions who behave sincerely, would not have the beliefs, attitudes, nor display the behaviours outlined above. This is particularly true for an institution that proclaims Christ as its founder. As Sipe (2008) revealed, the stakes surrounding exposure of clergy misconduct and general sexual activity are very threatening for the RCC. As such, the general response is to neutralise those reporting CSMAA. Accordingly, it would be very difficult for officials dealing with CSMAA to define it more as those who have researched this issue believe it should be defined: a form of professional misconduct, a betrayal of trust, abuse of power, abandonment of fiduciary duty (Fortune 1989, pp. 37, 42, 101; Rutter 1989, pp. 27–28, 205; Lebacqz and Barton 1991, pp. 106–12; Peterson 1992, pp. 178–79; Flynn 2003, pp. 17–27; Kennedy 2009, pp. 4–5; Byrne 2010, pp. 9–16, 51–52; Maltese Ecclesiastical Province 2014, pp. 13–14; Tschan 2014, pp. 46–48; de Weger 2020, pp. 66–70). Without such a definition there simply is no real victim or offender, or someone harmed and someone responsible for that harm (Rutter 1989, p. 23; Flynn 2003, p. 11). To fully acknowledge a more accurate definition, as all the authors just cited would call it, would be to instantly require justice for the offended, action against the offender, and even greater institutional transparency. Systemically, it would also require an actual and genuine reformation of backstage realities in order for these to sincerely synchronise with frontstage proclamations of compassion and justice.

## 3. Conclusions

Past and current clergy culture militates against change and maintains an inadequate response to CSMAA. Normalisation or minimisation of clergy sexual activity renders complaints of CSMAA somewhat unimportant. Some even suggest there is an element of puritanism, and a general lack of compassion for clergy who have to cope with a great deal due to celibacy, or who may have entered the clerical life too young[12]. At present, it still appears that clericalist self-beliefs about their believed-to-be noble quest to celibacy, if they accept that quest, or their own personal interpretations of chastity being unnecessary for celibates, supersede victims' expressions of their abuse. As a result, those already harmed by abuse are re-harmed, this time by the institution they love/d (Pooler and Barros-Lane 2022, p. 6). Much of the blame for the inaction or even the beliefs surrounding CSMAA and towards victims thereof can be traced back in the undertow of the culture of secrecy and fear exposed by such as (Sipe 2008). There is still little evidence to suggest that past processes such as Towards Healing, have sincerely advanced in new companies such as the Australian Conference of Bishops run and owned Australia Catholic Safeguarding Ltd. The entrenched culture described in this article is still blocking any meaningful, practical, and healing action from occurring when victims report their CSMAA. Within that culture, *because* of that culture, secondary deviance, and abuse/misconduct recidivism, proliferate. In turn, such misconduct then requires further deviancy in the need to cover-up. These deviant and sometimes criminal responses themselves then become expected and normalised.

Shupe maintained that powerful institutions are desperate to neutralise scandals within their structures secretly with as little cost as possible at first, but eventually at any cost it seems. At present, the RCC is no exception as many news stories and lawsuits including those discussed in this article (e.g., Lynne) have revealed (see also de Weger 2020, Appendix 1; Hainsworth 2020; The Pillar 2022). As such, the RCC culture clearly needs reformation.

All the participants of the studies underpinning this article stated that they really wanted to participate. Why? They hoped they could facilitate change by telling their stories and sharing them through such as this article; they really hoped there were people willing to listen. It is hoped that this article has helped them achieve their desire. They are brave and brilliant women and men, and we should be deeply grateful for their courage, conviction, persistence, and knowledge.

**Funding:** Both my Masters degree and Ph.D. were paid for as per the Commonwealth Government of Australia funding arrangements for higher degree research. I also received a university QUT scholarship for the final 2 years of my Ph.D.

**Institutional Review Board Statement:** Both the Masters research project and the Ph.D. study underwent in-depth scrutiny by the University Human Research Ethics Committee (Queensland University of Technology). Ethics approval was granted for both studies. Masters ethics approval number: 1300000401. Ph.D. ethics approval number: 1800001033.

**Informed Consent Statement:** Informed consent was obtained from all subjects involved in the study. As part of the informed consent agreement, it was made clear that the data received during the research projects would be used to inform the Church and other interested parties and would be used as a foundation for academic articles in the future.

**Data Availability Statement:** Data for both studies (Masters survey, Ph.D. interview transcripts) are not publicly available but may be available by contacting me directly at stephen.deweger@qut.edu.au.

**Acknowledgments:** I wish to acknowledge the support of Doris Reisinger, my fellow researcher into this major issue within the Catholic Church. I also would like to thank the MDPI team for inviting me to present this article here. It means a great deal to me personally and in regard to all the participants in my research who wanted their stories told so as to make a difference in the Church and for other survivors.

**Conflicts of Interest:** The author declares no conflict of interest.

## Notes

¹    For the underlying methodology and methodological approach for this study see the chapter, 'Research Design' (pp. 129–50).

²    For the underlying research design for this study see the chapters, 'Theoretical Framework' and 'Methodology' (pp. 60–70).

³    Unfortunately, more complete descriptions both of survivors' experiences of abuse and of reporting cannot be given here. For these, please refer to the de Weger (2020) on which this article is predominantly based. These more complete accounts give a much greater context and support to the findings in this article.

⁴    Regarding Church Militant for whom Niles works: It has been particularly interesting and somewhat troubling that for the most, it has been only such often 'gay-focused' Catholic organisations or groups that have been the only ones to deeply investigate the issues discussed in this article. Regardless, though of their particular leanings and biases, I give all sources a hearing. It is hard to fault the actual facts presented in this horror story which depicts exactly what Sipe, by no means a conservative, tried to warn about a decade ago. However, it is a breath of fresh air that this journal, *Religions*, has created this special edition dealing with CSMAA—doing so will present a more balanced account not the least because it includes CSMAA involving women as well.

⁵    The Melbourne Response was developed just before the release of the Towards Healing protocol, by the then Archbishop Pell. Many believe this was in response after stern words from the then Premier of Melbourne (Gordon and Armitage 2014). The Melbourne Response only dealt with diocesan clergy. However, many found Pell's actions puzzling because Towards Healing had already been developed by such as the later ostracised Bishop Geoffrey Robinson, along with lay church leaders and academics, and was almost ready to be implemented. However, as explained by one of the lay people involved, Towards Healing soon became an arm of Catholic Church Insurances and legal firms.*We went into what we called a survivor mission. We started writing a draft protocol for them. And of course, they adopted nothing like what we put forward. Our draft was far more victim oriented (Private correspondence).*According to this contact, other RCC leaders became more concerned with controlling what the public heard and saw when it came to clergy sexual abuse and took over the control.

⁶    At the time of the study on which this article is based, the RCC in Australia was dismantling Towards Healing and The Melbourne Response and formulating a new body to oversee a more national approach to protection policies known as Australian Catholic Safeguarding Ltd. (Australian Catholic Safeguarding Ltd. 2021). There is no longer any national formal process such as Towards Healing, through which victims/survivors can seek compassion and justice.

⁷    As a very recent example of such neutralisation tactics in action, where victim, whistle-blower and media are intimidated, see the discussion of the Cardinal O'Brien case between a survivor's (Mr. Brian Devlin) literary account of this case in *Cardinal Sin: Challenging Power Abuse in the Catholic Church,* and the reporter (Ms. Catherine Devaney) who walked with him through the whole process (Root and Branch 2022). Unfortunately, at the time of writing there was no copy of this presentation, nor its transcript.

⁸    At first all the headlines and news stories about the *New Book VI* mentioned amazing new changes that now included abuse of 'adults', e.g., (ABC News 2021); (Sky News 2021); (The New York Times 2021). Once it became clear that the new additions to Book VI were not that new (regarding adults) and that the term 'adult' was still highly ambiguous (it was not even used once), the headlines curiously started to omit any references to 'adults'. The issue of a still existing lack of clarity as to what type of adult could be definitionally abused was the focus of a seminar held by St Paul's University, Ottawa, Canada and presented by Msgr John Renken (St Paul University 2022). However, there is no recording of this event.

⁹    While most of these quotes have been presented earlier, it is worth presenting them again because of their central significance to this tactic and this study.

¹⁰    Feminist Standpoint Theory (FST) was the theoretical underpinning of the research approach of both de Weger (2016), and de Weger (2020). See de Weger (2016, pp. 60–66) particularly for a more detailed discussion of FST and why it was wholly appropriate for these studies.

¹¹    This document was presented as part of the interview.

¹²    See comment "NMichael Kellyame | 8 June 2017" (Fr Michael Kelly SJ), in de Weger (2017).On further inspection, his original comment has been removed and replaced with a much more supportive one if it is indeed 'NMichael Kellyame' writing. This is the original one:*I regret to say I cannot join the applause for Mr. de Weger's contribution. There is a fatal conflation between what is considered normal sexual behavior—falling in love and developing a relationship—with something that is criminal and pathological—the sexual abuse of minors. They are different matters. For reasons Mr. de Weger gives, the reporting of the latter only really started to be done relatively recently. But I know for sure and certain it happened and was acted on 70 and 80 years ago. I know that from victims who 've told me their stories and from what I know of the actions of Jesuit superiors in the 1930s,'40s,'50s and '60s.The assertions about adult misbehaviour also neglects a basic fact which I have known for 50 years, 47 of them in the Jesuits. Many of those who ambitiously committed to a chaste and celibate life should never have and did so for unsustainable reasons like parental or peer pressure for example (my emphasis). That falling in love helped them to grow up should be seen as something positive. Something more about cultural and ecclesial context would enhance your argument, Mr. de Weger.* Michael Kelly 8 June 2017.Eureka Street is a Jesuit magazine and Fr Michael Kelly SJ, one of the original creators thereof.

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
