# Peer review of "Insincerity, Secrecy, Neutralisation, Harm: Reporting Clergy Sexual Misconduct against Adults—A Survivor-Based Analysis"

_religions, doi:10.3390/rel13040309_

Round 1

Reviewer 1 Report

This is an excellent paper.  Overlaying the findings on top of Shupe's identifications of how the church creates barriers in its response is outstanding. This paper will make a significant and noteworthy contribution to the literature on clergy sexual abuse of adults. This paper is timely and needed. I applaud the author(s) on such a relevant and useful study. The findings are described exceptionally well.  However, I do have a few suggestions for improvements that can be made. 1). I would like the author to take a bit more time for methodology and describe how the 9 people were recruited.  Was there an interview guide used?  Are there some additional questions that were used to guide the interviews not mentioned in the paper? and how were the data analyzed?  2). Pooler and Barros-Lane (2022) just recently had a paper published and they use the term Clergy Perpetrated Sexual Abuse of Adults.  It is not necessary that the author use this term, but it would be helpful to note that there are other terms being explored in the literature (as this is still an understudied field).  In addition in that same paper is mention that how the institutional church responds to reports of abuse is often worse than the abuse itself.  I think it would further boost the author's findings and explanation of how exactly this plays out specifically in the RCC context to note how this current paper under review really explains and explores that phenomena mentioned in Pooler and Barros-Lane's work in greater depth. 

3) Below I have a couple of other minor edits.

Line 355 I would like the author to better explain legalism.  As a reader I'm unclear exactly how that is being used here and exactly what it means.

Line 813 -- for most would work better than "for the most"

Again well done! It was a pleasure reading and reviewing this paper. 

Reviewer 2 Report

The methodology of the study on which this article is based needs to be elaborated. In particular, explanation is required of how the participants in the study were identified, selected and recruited? What means were used to protect their identity and ensure anonymity and confidentiality? What instruments were used to obtain the data which feature in the discussion and analysis, and on which the conclusions of the study are based? In the case of surveys and/or interviews, what questions were asked or prompts employed?

The terms “redacted/de-identified for review purposes” appear at many points throughout the version of the article that this reviewer has received. I understand that it is the intention that these terms be removed once the review process is completed, and that the title of an already available work will replace these terms in the published article. Thus, by withdrawing this material now from the scrutiny of the reviewer can only be to protect the author’s identity. This, however, should not preclude here the author’s giving some brief account of the methodology followed in the earlier work.

The Abstract advises that: “The foundational study for this article asked: how do survivors of clergy sexual misconduct against adults (CSMAA) in the Roman Catholic Church (RCC) describe and understand their experiences of reporting that misconduct to Roman Catholic Church authorities?”

Then, at the beginning of the second paragraph of the Introduction, one reads:

Reporting clergy sexual misconduct against adults to Roman Catholic Church authorities: An analysis of survivor perspectives (redacted/de-identified for review purposes) was an Australian-based study undertaken because there was a very large gap in the literature and research regarding CSMAA, and a complete vacuum of research into the reporting thereof.

The reader assumes that this is “the foundational study” referred to in the Abstract.

As the same paragraph of the Introduction continues:

In particular, the actual voices of victims/survivors of CSMAA, had rarely been brought to the fore. This project was also undertaken because it became clear in redacted/de-identified for review purposes [sic] that many of the participants who had reported their abuses to RCC officials, had experienced negative responses but those responses were not able to be dealt with at the time.

The reader might still assume that “[t]his project”, in the second sentence above, refers again to “the foundational study”. It comes as something of a surprise, therefore, to read that:

The overall conclusion of both studies [reviewer’s highlight added] was that official church responses were deeply flawed due to erroneous beliefs about CSMAA, which in turn influenced the attitudes and behaviours towards survivors.

At this point the reader becomes aware that “the foundational study” and “this project” are two related but separate studies. Even though the questions posed above about methodology may already have been answered in “the foundational study”, a summary of these considerations needs to be incorporated into the present article. It is essential that the reader have some sense of the process that has led to the acquisition of the data analysed and conclusions reached. This will enable readers to appreciate the impact of the abundant quotations from participants and to place them in relation to the 11-point thesis of “tactics and modes of neutralization” adapted by Sipe from Shupe, and evaluate the effectiveness of this scheme as applied to these data.

Note (i) is quite puzzling. This reader is asked to: “please refer to the (redacted/de-identified for review purposes) on which this article is predominantly based. These more complete accounts give a much greater context and support to the findings in this article.”

This is to put the present reader reviewing this article before an impossible task. Without access to “(redacted/de-identified for review purposes)”, this reader is deprived of the predominant basis of “this article”, as well as “context” and “support” for its findings.

Concerning matters of detail:

42        a global one

46        “at the time.” Which time? Make clear what this relates to.

51        “… action against clergy sexually active and misconducting clergy.”

Recommended: “against sexually active and misconducting clergy.”

55-7                             “Much of this insincerity could be traced back to the RCC’s own ambivalence about clergy sexual activity and misconduct, not to mention being caught off guard about that ambivalence (redacted/de-identified for review purposes).”

A prime example of where the reader needs to know something more specific about what has been (redacted/de-identified for review purposes).

101      “within powerful”. Word omitted after “powerful”.

101-2   “we should be under no delusions”

            Recommended: under no illusion

111      “power inevitably corrupt”

The quotation from Lord Acton is: “power tends to corrupt, absolute power corrupts absolutely.”

123      Reference to Daniel should be to 2:33.

140      “ingenuous” to be replaced by “disingenuous”

141                  “a confusing duality of intent accompanied by an ingenuous spirit of compassion and justice on behalf of the officials to whom survivors went to report their CSMAA.”

Replace “on behalf of” with “on the part of”

170                  “to how they were being responded.”

                        Recommended: “to how they were being responded to.”

175-7               “In short, all participants in redacted/de-identified for review purposes and many in redacted/de-identified for review purposes felt deeply harmed by their reporting and/or disclosure experience.” [Emphasis added by reviewer]

This makes it difficult for the reviewer to assess the evidence for this assertion regarding “all participants”.

179-80                         “other perhaps more accurate ‘backstage’ beliefs, attitudes, and other perhaps more accurate ‘backstage’ beliefs, attitudes, and behaviours held by officials dealing with CSMAA.”

                                    How is one to judge whether the ‘backstage’ factors are “perhaps more accurate”?

185-6                           “a serious issue of professional abuse enabled by steeply imbalance power contexts.”

                                    Recommended: heavily unbalanced.

188                              Recommended: this present article …

216                              “was developed” to become “were developed”

285                              “to be inert” to become “to remain inert”

318-23                         Did Vince in fact go to the police? For all the “buck-passing” involved on the part of church figures, this direct advice from Bishop A could be read to some extent positively, unless it were clear that the Bishop was obfuscating, knowing full well that Vince would not take up his recommendation.

390                                                      “that clarity is also still very obscure.”

                                                            Recommended: Even with this clarification, the matter is still very obscure.

413                              Shouldn’t this be “the remaining five”.

“Four of the nine participants” are mentioned in the paragraph immediately before the long quotation.

559                              “Ingenuousness” to become “disingenousness.”

681                              “MR” to become “the Melbourne Response”.

849                              “constructed” to become “construed”.

980-81                         “Sequalae of harms” to become “sequelae of harms”.

1001                            “diocese” to become “dioceses”.

1089                                                    write “the behaviours”

1120                                                    “in such new ‘companies’ such as” becomes

“in new ‘companies’ such as …”

  1. 25, n. iii, 2nd-3rd last line Sentence beginning with “Furthermore” lacks a main verb.
  2. 26, n. x Typographical error with “Michael Kelly’s” name

Reviewer 3 Report

  1. The article has the merit of highlighting the neutralization tactics used by the Catholics clergy in situations of abuse.
  2. The article may be a manifesto urging the assumption  of deviations and redefinition of  Catholic clergy values.       

Author Response

No changes were recommended.

A sincere thank you to Reviewer 3.